# Working Space Creation in Transoral Thyroidectomy: Pearls and Pitfalls

**DOI:** 10.3390/cancers14041031

**Published:** 2022-02-17

**Authors:** Tsung-Jung Liang, I-Shu Chen, Shiuh-Inn Liu

**Affiliations:** 1Division of General Surgery, Department of Surgery, Kaohsiung Veterans General Hospital, Kaohsiung City 81362, Taiwan; tjliangmd@gmail.com (T.-J.L.); nugaticc@gmail.com (I.-S.C.); 2School of Medicine, National Yang Ming Chiao Tung University, Taipei 11221, Taiwan

**Keywords:** transoral endoscopic thyroidectomy, parathyroidectomy, robotic, working space, platysma muscle, oral vestibule

## Abstract

**Simple Summary:**

Transoral thyroidectomy accesses the thyroid gland through three incisions in the oral vestibule. The cosmetic outcome was excellent since no scar was observed on the body surface. However, it is challenging to create a working space using this new approach. Unconventional but severe complications can also occur. Our review summarizes the tips regarding working space creation in transoral thyroidectomy and tricks for preventing complications.

**Abstract:**

Transoral thyroidectomy is a novel technique that uses three small incisions hidden in the oral vestibule to remove the thyroid gland. It provides excellent cosmetic results and outcomes comparable to the open approach. One of the main obstacles for this technique is the creation of a working space from the lip and chin to the neck. The anatomy of the perioral region and the top-down surgical view are both unfamiliar to general surgeons. As a result, inadequate manipulation might easily occur and would lead to several unconventional complications, such as mental nerve injury, carbon dioxide embolism, and skin perforation, which are rarely observed in open surgery. Herein, we summarize the basic concepts, techniques, and rationales behind working space creation in transoral thyroidectomy to assist surgeons in obtaining an adequate surgical field while eliminating preventable complications.

## 1. Introduction

Transoral thyroidectomy via the vestibular approach leaves no visible scar on the body surface while adhering to the minimally invasive concept by dissecting far smaller areas than other remote access thyroidectomies, such as transaxillary, retroauricular, or bilateral axillo-breast approaches. Growing evidence shows that transoral thyroidectomy has equivalent and even superior surgical outcomes compared with the open approach and other extracervical accesses [1,2,3,4]. Therefore, this procedure became popular worldwide [5] soon after Dr. Anuwong published his initial experience with the first 60 human cases in 2016 [6].

Nevertheless, the introduction of a new, novel technique will inevitably accompany some inherited complications that are unique and unfamiliar to surgeons. In transoral thyroidectomy, developing a working space through three small incisions in the oral vestibule is the core feature of this procedure and is the most challenging part for beginner surgeons who are trying to learn this technique. Many complications that occur during this step are rare or unprecedented in open thyroidectomy, such as carbon dioxide (CO_2_) embolism, mental nerve injury, drooling, and skin flap perforation [7,8].

This review therefore aims to present the current approach and evidence on each step of working space creation in transoral thyroidectomy. We also demonstrate tips and tricks on how to create a suitable space and prevent devastating complications. To the best of our knowledge, a comprehensive review specifically focusing on working space creation in transoral thyroidectomy has not yet been reported.

## 2. Basic Concepts

### 2.1. Patient Selection

The patient selection for transoral thyroidectomy is similar to that for other remote access thyroidectomy, including the inclusion of those with benign lesions ≤5 cm and well-differentiated carcinoma ≤2 cm [5]. Larger tumors may be difficult to manipulate in the narrow working space and complete specimen retrieval without capsule disruption may not be feasible through the small oral incision [9]. The integrity of the specimen is very important when operating on patients with suspected malignancy because tissue fragmentation would interfere with the pathological examination [10]. Some surgeons suggest extracting the large specimen with an additional trocar in the axilla [11].

In the top-down view of transoral thyroidectomy, the lymph nodes at level VI can be clearly identified and central compartment dissection can be performed. However, lateral neck lymph node dissection, especially at level II, is extremely challenging via the transoral approach [12,13]. Thus, patients with preoperative imaging evidence of lateral neck involvement may be excluded or operated only by experienced surgeons [14]. Other exclusion criteria contain ongoing oral infection and poorly differentiated cancer. Operation in patients with prior neck surgery, previous head and neck irradiation, or extensive thyroiditis are technically demanding and are relatively contraindicated [15,16].

### 2.2. Subplatysmal Space

The subplatysmal space is a potential plane with a discrete boundary (above: platysma muscle; below: anterior jugular vein/superficial surface of the deep cervical fascia) [17]. It can be expanded to a working space by the following three steps: hydrodissection, blunt dissection, and sharp dissection.

First, expansion fluid is injected into the subplatysmal area for hydrodissection. It aims to make the space thicker for easier access and separate the platysma muscle from the underneath strap muscle. Second, blunt dissection with a straight rod or a Hegar dilator is performed to raise the skin flap. Finally, sharp dissection with an energy device or electrocautery is performed to divide the residual attachment and complete the working space creation [18].

### 2.3. Patient Position

In thyroidectomy, the patient is placed in the neck extension position to lift the thyroid gland up for better exposure. Neck positioning can be performed with a shoulder roll or a thyroid pillow (Figure 1). In the early era of transoral thyroidectomy, extensive neck hyperextension was preferred by some surgeons to make the chin, thyroid cartilage, and sternal notch in the same plane [19]. It is presumed to be better for manipulation using a straight, non-angulated endoscopic instrument. Sometimes, the head section of the operating table must tilt further down to achieve this alignment [20].

However, excessive neck extension may increase skin tension and compromise the elasticity of the working space [21]. The skin flap will be too tense to be raised sufficiently high for instrument manipulation despite CO_2_ insufflation and additional hanging sutures for external lifting.

Currently, most surgeons position the patient with slight neck extension, similar to open thyroidectomy, and keep the neck skin slightly loose [5]. In this way, the skin flap can be fully distended after CO_2_ insufflation, and a larger space can be obtained. It may also decrease postoperative cervical pain and the risk of spinal cord injury [22,23,24].

In addition, a 15–30 degree Trendelenburg position can provide better access to the neck [22].

## 3. Operative Techniques

Table 1 summarizes the key points at each step of working space creation in transoral thyroidectomy.

### 3.1. Incision

#### 3.1.1. Mental Nerve

The mental nerve is a pure sensory nerve that is derived from the inferior alveolar nerve. After emerging from the mental foramen of the mandible, the mental nerve divides into several branches that innervate the gingiva, lower lip, and chin [25].

The key point of the incision design in transoral thyroidectomy is to avoid injury to the mental nerve, which might impair the sensation of the lower lip and chin [26]. In keeping with anatomical studies of the mental nerve, we adopted the concept of a safety triangle [11,27,28]. A 1.5–2-cm inverted-V shape incision was made just above the labial frenulum in the central location to avoid injury to the medial branch of the mental nerve (Figure 2) [11]. Another option is to make the central incision vertically [29]. Peng et al. [30] suggested wide mucosal dissection to fully expose the ramifications of the mental nerve and determine the optimal site for trocar insertion. Yeh et al. [31] used computed tomography simulation and 3D printing techniques to create a module that marks the location of the mental foramen and thus guides the surgeons to avoid mental nerve injury.

The other two lateral incisions (0.5 cm) were made close to the inner edge of the bilateral mouth commissure [5,32].

#### 3.1.2. Mentalis Muscle

The mentalis is a pair of dome-shaped muscles located at the central chin that acts as the major elevator of the lower lip. It originates from the incisive fossa of the mandible and inserts into the skin of the chin [33,34]. The mentalis is referred to as the pouting muscle because the lower lip is raised and protruded out when it contracts. The contraction of the mentalis muscle also wrinkles the skin of the chin and is essential for the facial expression of doubt and contempt.

Injury to the mentalis muscle may impair the vertical support of the lower lip, which leads to drooling. The lower central incisors are visible at the resting position and might impair denture stability [33,34].

After incising the oral mucosa, the central incision was further dissected down to the lower edge of the mandible with an electrocautery or Kelly clamp. The mentalis muscle might be cut or penetrated during this process [18,35,36,37,38]. This injury might lead to uncoordinated movement of the lower face during pronunciation or smiling [39,40]. In addition, skin perforation might occur due to heat transmission from electrocautery [41] or direct penetration by applying excessive force using a Kelly clamp (Figure 3). Initial occult injury in such a thin flap might also lead to skin dimpling that does not resolve over time and affects the patient’s aesthetic outcome [42]. Great care should be taken to check the thickness of the chin flap at this stage.

Hence, some surgeons prefer deeper dissection at the periosteum level of the mandible (Figure 4) [16,39,43]. Dissection through this layer goes underneath the mentalis muscle, and the muscle can be mostly preserved, decreasing the associated risk of muscle bleeding [43]. In addition, obtaining a thicker skin flap lowers the risk of flap perforation. Anatomic studies suggest that the periosteal approach allows easier access to the subplatysmal space [44].

### 3.2. Hydrodissection

#### 3.2.1. Expansion Fluid

The expansion fluid comprises epinephrine with normal saline (1:200,000–500,000) for vessel constriction and elimination of bleeding [18,45]. In bilateral axillo-breast approach thyroidectomy, the addition of ropivacaine was found to alleviate postoperative pain [46]. This may also be considered in transoral thyroidectomies.

#### 3.2.2. Injection Tool

In a pilot study by Dr. Anuwong, a Veress needle was used to inject the fluid via three mucosal incisions [6]. Since the Veress needle is thick and firm, it is easier to manipulate the needle and deliver the fluid to expand the desired subplatysmal space and infiltrate the trocar tract in advance. However, the chance of vessel injury may be higher with a thick needle even though the syringe plunger is pulled back every time for checkup before fluid injection. In addition, tracheal perforation by a Veress needle puncture has been reported [38]. Therefore, some surgeons prefer using needles with a small caliber (e.g., 22-gauge spinal needle) for hydrodissection [45,47]. Thin needles are difficult to control when approaching deeper spaces. Additional percutaneous puncture may be required to infiltrate the lower neck. Nevertheless, these needle holes were relatively small and negligible within 1–2 days postoperatively.

#### 3.2.3. Injection Volume and Location

The volume of fluid injected usually ranges from 30–60 mL and depends on several factors [18,38,45]. On the patient’s side, young patients with dense tissue require more fluid to expand the tense space. More fluid is also injected in lean patients with a thin skin flap to prevent flap perforation. On the surgeon’s side, more fluid would spread the initial narrow space wider for easier dissection. However, more fluid contributes to more vapor formation when an energy device or electrocautery is used in the subsequent stage of sharp dissection. The smoke obscures the surgical field and needs to be removed by suction, which can disrupt the surgery.

Expansion fluid can infiltrate the entire space that is planned to be dissected. In contrast, for surgeons who prefer less smoky operative fields, hydrodissection can be limited to some key structures that only have thin tissue overlaid, such as thyroid cartilage, cricoid cartilage, and tract for lateral trocar insertion. Less fluid retention may also be associated with less postoperative regional swelling, which heals within a few days.

### 3.3. Blunt Dissection

#### 3.3.1. Dissector

The commonly used dissectors in transoral thyroidectomy are blunt tip straight rods [18,48] or Hegar dilator [16,49]. Hegar dilator is a slightly curved metal rod, which may be easier to apply when passing through the concave surface between the chin and neck. Hegar dilators also have various diameters, which allow for gradual dilatation [35].

Balloon dilatation with a balloon in the Foley catheter [50], cuff in the endotracheal tube [51], and commercialized balloon dilator device [47,52] are also alternative options used by some surgeons. With the balloon inflation pressure, it is easier to create a homogenous space with a clear tissue plane and achieve good hemostasis [50]. In addition, an endoscopic camera can be inserted inside the transparent endotracheal tube or dissecting balloon for real-time inspection of the dissection process [51,52].

#### 3.3.2. Dissection Technique

Blunt dissection is initiated by passing the Kelly clamp through the central incision. The clamp was gradually advanced and spread out until the level of the thyroid cartilage. During this process, great care should be taken to check the tip location of the Kelly clamp to avoid direct flap penetration (Figure 3).

Next, the blunt tip straight rod or Hegar dilator was used to create the subplatysmal tunnel and elevate the skin flap. The number of tunneling can be multiple as fan-shaped [48,49] to lift the whole skin flap or just a few strokes to open an initial space for sharp dissection.

In our institute, tunneling is only performed once to make an initial tract that can accommodate the Foley catheter for subsequent balloon dilatation [50].

Kim et al. [53] proposed the use of a special curved tunneler for blunt dissection of the upper space (cranial side) as high as 2 cm above the bilateral superior pole of the thyroid. In this way, lateral port insertion would be easier and their ranges of motion would also increase.

#### 3.3.3. Complication and Prevention

It takes some practice to blindly enter the correct subplatysmal plane through blunt dissection. If the dissecting plane is too shallow, skin dimpling and contracture may occur [54]. Bleeding may occur if the plane is too deep and enters the strap muscle, which makes further dissection difficult and results in postoperative bruising and ecchymosis.

Usually, bleeding at this stage is self-limited or can be managed endoscopically after trocar insertion and CO_2_ insufflation. However, two cases of CO_2_ embolism that occur immediately after gas insufflation have been reported in the literature [42,55]. Both patients developed hypotension and asystole and received cardiopulmonary resuscitation. It has been postulated that CO_2_ enters the blood stream through vessels lacerated during blunt dissection. After experiencing this serious complication, Hong et al. [42] routinely applied external compression for 1–3 min to achieve hemostasis before initiating CO_2_ insufflation, and the use of blunt dissection was minimized to lower the risk of vessel injury.

For those who use balloon dilatation, the inherent compression pressure of the expanding balloon may be beneficial for hemostasis [50]. Small venous oozing can be separated by pressure without further management.

Long et al. [35] reported a case of airway injury at the level of the thyroid notch in a female patient who underwent transoral endoscopic right thyroidectomy. The trachea was pierced by a small-caliber, narrow-headed Hegar dilator during blunt dissection. Despite this, the air leak did not interfere with general anesthesia, and the operation was completed endoscopically. The perforation site was closed with 3-O sutures and reinforced by muscle coverage plus fibrin sealant application. The patient’s postoperative course was uneventful. To prevent this adverse event, the author suggested using a more blunt and larger Hegar dilator (diameter > 10 mm) for the initial dilatation [35].

### 3.4. Trocar Insertion

#### 3.4.1. Trocar Selection

In transoral thyroidectomy, the classic setting is a 10–12 mm trocar in the center with two 5 mm trocar at each side near the oral commissure. Since the distances among the three trocars are relatively close, it would be better to use a trocar with a small head to avoid inter-trocar collision [56]. Furthermore, James et al. [16] use a 5 mm trocar in the middle instead of the original 10 mm one to perform most of their transoral surgeries. It was not until the end of the surgery that the central tract was dilated to facilitate specimen removal. This modification would lower the chance of collision between trocars and is associated with less local induration because less tissue is damaged [16].

Trocar dislodgement occurs frequently because only a small portion of the cannula can be inserted into the limited narrow space. Therefore, using a short cannula with a threaded design may hold the cannula from moving and prevent slippage during surgery. The 5 mm lateral trocars should have a side port for gas venting and smoke evacuation, thus maintaining the visibility of the surgical field [20].

#### 3.4.2. Position of Trocar

To avoid inter-trocar collision, the venting side port should face outside, and the heads of the cannulas should not be at the same level to decrease collision (Figure 5).

Prior to lateral port insertion, the imaginary insertion tract should be filled with expansion fluid through hydrodissection, especially at the jaw line, where the skin is thinnest, to avoid skin penetration [57].

To facilitate thyroid upper pole dissection, the bilateral working ports should be inserted toward the upper tip of the thyroid lobe on each side, in a parallel manner rather than inserted toward the central part of the cricoid cartilage [53]. The angle between the central and lateral trocars should not exceed 45° to prevent compression injury of the mental nerve [58].

The tip of the camera port should ideally be at the level of the inferior jaw edge for better visualization [22]. Deeper insertion of the camera port can compromise the visual field, especially during thyroid superior pole dissection [22].

### 3.5. Sharp Dissection

Prior to sharp dissection, one or several hanging sutures were placed in the midline below the tip of the central trocar to lift the space (Figure 5). An energy device or hook cautery was applied to create the working space. The active blade of the energy device should always be kept away from vital structures, such as the trachea. The desired boundary is superior to the thyroid cartilage, inferior to the sternal notch, and lateral to the medial border of the sternocleidomastoid muscle [18].

#### 3.5.1. Subplatysmal versus Subfascial

In endoscopic thyroidectomy, the subplatysmal dissecting plane should be close to the strap muscle rather than close to the platysmal muscle to obtain a thicker skin flap [16]. The thicker the flap, the lesser the chance of skin burn, perforation, dimpling, and skin contracture [54].

Some surgeons prefer to perform the dissection even deeper, below the anterior fascial layer of the strap muscle, the so-called subfascial dissection [17,59,60]. Dissection through this plane will lift the anterior jugular vein on the roof, thus reducing the chance of venous laceration. This dissection approach also preserves anterior neck sensation by protecting the transverse cervical nerve, which lies in the superficial layer of the deep cervical fascia and is above the dissection plane [17]. The thicker skin flap and less sensory nerve injury may be the reasons why fewer postoperative swallowing discomforts were observed in patients who underwent subfascial dissection than in those who underwent subplatysmal dissection [17,59].

Furthermore, the linea alba cervicalis is clearly seen after subfascial dissection, and the strap muscle can be easily separated in the midline.

#### 3.5.2. Anterior Jugular Vein Bleeding and CO_2_ Embolism

Bleeding in a narrow space is difficult to manage. The most common vessel injured at this stage was the anterior jugular vein. Once bleeding occurs, the bleeder should be seized with direct compression by an instrument on one hand, and another instrument should be used to achieve definite hemostasis, preferably with an energy device or clip.

Unable to identify the bleeder or at least temporarily stop the bleeding may pose the risk of letting CO_2_ enter the lacerated vein and subsequently lead to fatal CO_2_ embolism. Hypotension and even asystole have been reported in patients who develop CO_2_ embolism during transoral thyroidectomy and parathyroidectomy [55,61]. In such cases, CO_2_ insufflation should be terminated and 100% oxygen should be provided. Patients were placed in the left lateral decubitus and Trendelenburg position to preclude the gas emboli from occluding the right ventricular outflow tract [62].

In our institute, if bleeding cannot be controlled in a timely manner and CO_2_ embolism raises a concern, we would stop CO_2_ insufflation, place additional external hanging sutures to lift the space, and try to achieve hemostasis under gasless circumstances. Alternatively, open conversion remains the gold standard for uncontrolled bleeding during endoscopy.

## 4. Special Consideration

### 4.1. Male Patients

Male patients tend to have square jaws and very dense soft tissues in their chins, which are difficult to dissect [63]. In addition, the prominent thyroid cartilage impedes the instrument from crossing the midline and limits its range of motion [63]. The laryngeal prominence is also a danger zone for both blunt and sharp dissection because the overlying skin is very thin, and the risk of perforation is high.

### 4.2. Completion Thyroidectomy

Few studies have discussed completion thyroidectomy via the transoral vestibular approach [11,48,64]. The major concern is whether the working space can be created again via the same route. Most surgeons believe that repeated creation of the working space is easy and feasible if it is performed within 1–2 weeks following the initial thyroidectomy because the adhesion formation is not mature yet [18,64]. After 2 weeks, tight adhesion may pose difficulty in opening the space again. Wu et al. [65] proposed creating a working space through a different approach if completion thyroidectomy was performed beyond 2 weeks. Anuwong et al. [18] suggested postponing the surgery 6 months later if a repeated transoral approach is desired.

At present, only a small number of patients have been reported to undergo transoral completion thyroidectomy beyond 2 weeks after index surgery [11]. In our institute, 5 patients underwent completion thyroidectomy via the transoral vestibular approach. The time interval between the two surgeries exceeded 2 weeks in three patients, and there was no open conversion. Similar experience from the bilateral axillo-breast approach also revealed a 100% success rate in 33 consecutive patients who underwent completion thyroidectomy through the same route [66]. The mean interval between the two surgeries was 5.5 months in their study. We think that meticulous flap dissection at the first time and the use of anti-adhesive may be beneficial for reopening the same working space again. However, further studies are required to address this question.

### 4.3. Surgical Competency

Previous studies have demonstrated a short learning curve for transoral thyroidectomy, with fewer than 20 cases required for skill acquisition [1,67,68]. The learning curve for transoral thyroidectomy is shorter than that for other endoscopic approaches, such as the transaxillary and retroauricular approaches [69,70,71]. The symmetric port placement and identical midline approach as open surgery may be beneficial for surgeons to rapidly gain proficiency in transoral thyroidectomy.

Nevertheless, rare but severe complications did occur in transoral thyroidectomy. Surgeons who are interested in using this novel technique should be well-prepared and receive in-depth training before working on their first case.

## 5. Postoperative Care and Evaluation

### 5.1. Pressure Dressing

After the operation, a pressure dressing was placed on the chin for one day [18]. An elastic bandage such as the jaw bra is another alternative, which may provide better compression [20]. Currently, there is no robust evidence on how long these compression garments should be used, and the benefits of reducing swelling, ecchymosis, and seroma prevention remain unclear.

### 5.2. Oral Wound Care

Postoperative oral hygiene was maintained with mouthwash after meals. Brushing teeth is feasible, but it can elicit pain [72]. Using a toothbrush with a smaller head or a cotton swab may make it easier to navigate the narrow space in the oral vestibule.

### 5.3. Skin

Bruising and ecchymosis are transient and usually resolute in a few days, such as regional swelling [54]. Skin dimpling may resolve after soft tissue remodeling but would also remain if scar contracture develops. Thermal injury may initially present as erythema but later turn into full-thickness skin necrosis, which requires excision and even local flap coverage [41,54].

### 5.4. Altered Sensation

Sensory alterations are common following transoral thyroidectomy. Liang et al. [8] reported that 72.5% of patients had sensory changes in the chin, followed by 52.9% in the lower lip, 33.3% in the upper neck, and 5.9% in the lower neck on the second day after surgery. Most sensory impairments are transient and resolves within 3 months [8].

While sensory changes in the lower lip and chin can be attributed to mental nerve injury, the sensation of the neck is not innervated by the mental nerve, and its alteration is more likely caused by damage that occurs during working space creation. Some patients would feel tightness along the surgical tract that clench their neck and make them unable to raise their heads [7,41]. This so-called pulling sensation is probably caused by fibrosis and adhesion formation of the skin flap. It may take up to six months for this sensation to disappear completely [41].

### 5.5. Motor Funcion and Mimmetic Expression

Motor function of the mimetic muscle may be impaired after transoral thyroidectomy, which affects facial expression of smile, talk, eat, and other motions that involve the coordination of the lower lip and chin [39]. Plausible injury mechanisms include regional tissue edema, tears of the mentalis muscle, and injury to the branch of the facial nerve [39].

Leakage of liquid or drooling is an irritating symptom that might be caused by multiple factors, including (1) sensory dysfunction, unnoticed liquid droplet on the lip; (2) motor dysfunction, mentalis muscle laceration; (3) discoordination between sensory and motor feedback loop [8,30,39]. Although drooling in most cases is self-limiting within one month, it significantly affects the patient’s quality of life [8].

Zheng et al. [39] reported that 23.1% of patients experienced abnormal motor function of the lower lip and chin in their early practice. The rate dropped to 2.4% after they modified their incision design to preserve the mentalis muscle and avoid mental nerve injury.

## 6. Conclusions

In this review, we summarize the basic concepts, techniques, and rationales behind working space creation in transoral thyroidectomy to assist surgeons in obtaining an adequate surgical field while eliminating preventable complications. The procedural recommendation is derived from currently available literature. Further exploration of the detailed technique is warranted to improve this surgical procedure for better performance.

## Figures and Tables

**Figure 1 cancers-14-01031-f001:**
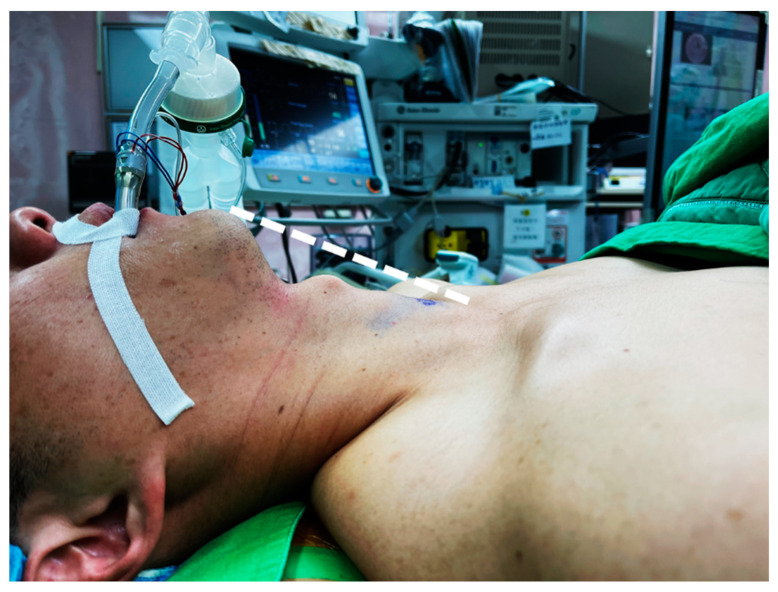
Patient was placed in slight neck extension. The loose skin can be lifted (dotted line) after CO_2_ insufflation and provide sufficient working space for instrumentation.

**Figure 2 cancers-14-01031-f002:**
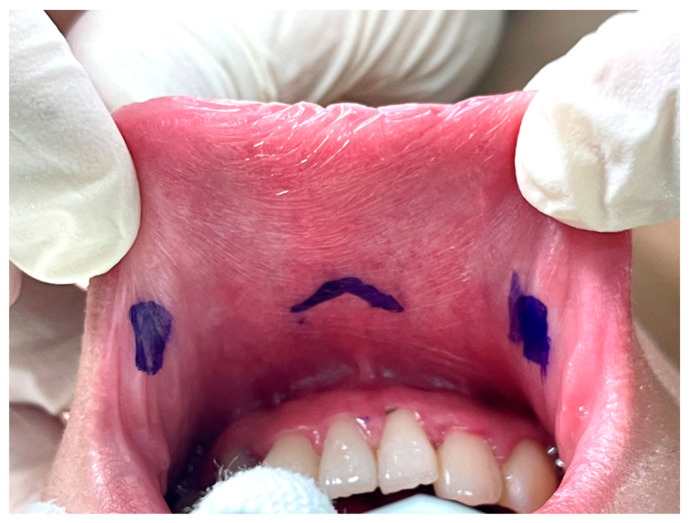
Vestibular incision design in transoral thyroidectomy.

**Figure 3 cancers-14-01031-f003:**
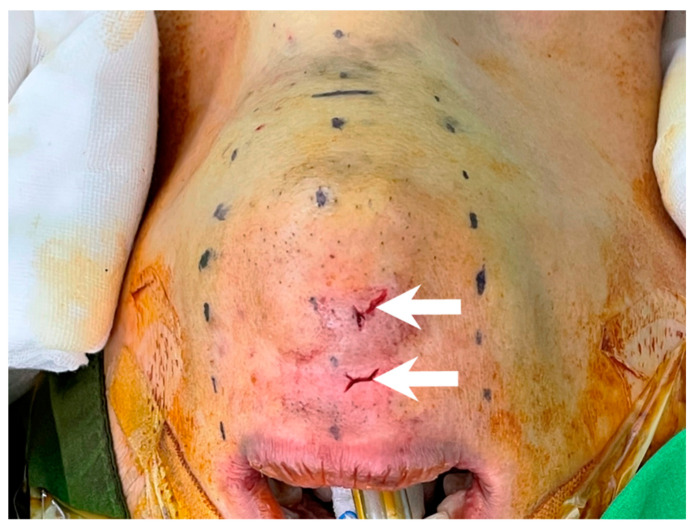
Chin perforation (arrow) by inadvertent use of Kelly clamp during central tract dissection.

**Figure 4 cancers-14-01031-f004:**
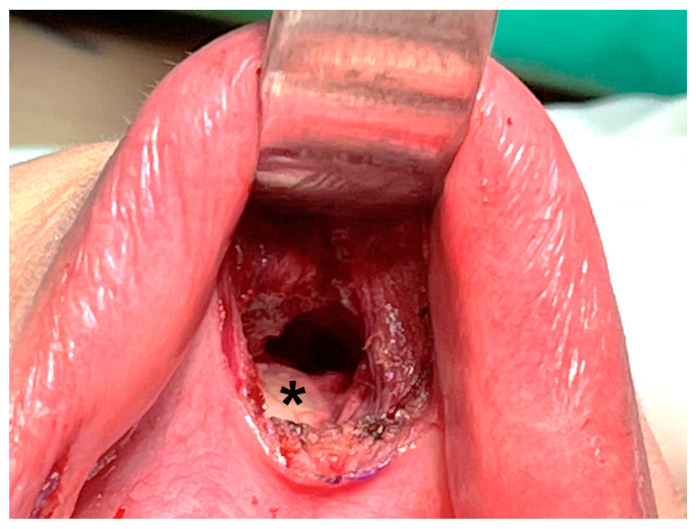
Dissection is performed deeper at the periosteal level (asterisk) of the mandible.

**Figure 5 cancers-14-01031-f005:**
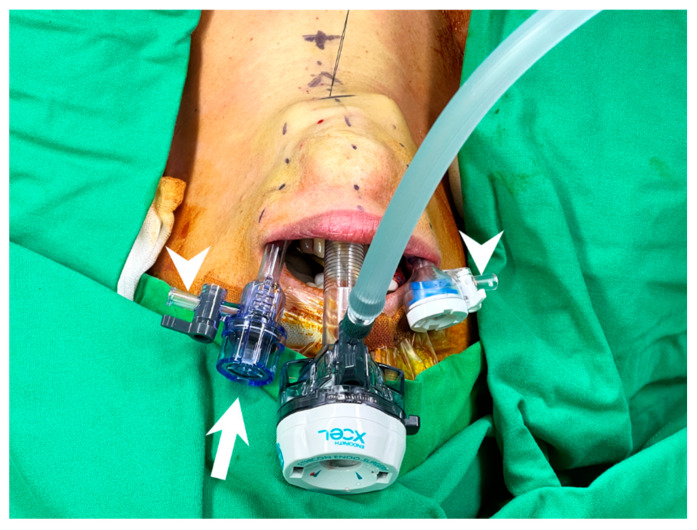
Trocar insertion in transoral thyroidectomy. Trocars with small head are preferred and the left lateral trocar should be inserted a bit deeper to avoid inter-trocar interference (arrow). The venting side port should be turned away from center and faced outside (arrowhead) to eliminate collision.

**Table 1 cancers-14-01031-t001:** A summary of key points for working space creation in transoral thyroidectomy.

Stage	Pearls	Pitfalls
Patient position	Position as open thyroidectomy, avoid excessive neck extension 15–30 degree Trendelenburg position	Tense skin and difficult to raise the skin flap Postoperative cervical pain spinal cord injury
Incision	Safety triangle concept for incision design Deeper dissection plane close to the periosteum of the mandible Cautious use of electrocautery and Kelly clamp	Mental nerve injury Mentalis muscle injury Chin perforation/skin dimpling
Hydrodissection	Cautious use of Veress needle for injection (alternative: spinal needle) More fluid injection for young patients with dense tissue, and structure with thin tissue overlaid (e.g., thyroid cartilage, tract for lateral trocar insertion) Avoid injecting large volume of expansion fluid	Vessel injury Tracheal perforation Excessive smoke that obscures the surgical field Regional swelling due to fluid retention
Blunt dissection	Limited use of blunt dissection (alternative: balloon dissector) External neck compression 1–3 min after blunt dissection for hemostasis Use thicker, blunted-tip dilator (diameter > 10 mm)	Bleeding CO_2_ embolism Skin perforation Tracheal laceration
Trocar insertion	Short trocar with threaded design to prevent slippage Use trocar with small head portion, insert in different depth, and turn the venting port away from center Avoid inserting camera port deeper than the jaw edge for better vision Insert lateral trocar toward bilateral upper thyroid rather than toward central cricoid cartilage for better superior pole dissection	Trocar dislodgement Trocar collision Poor visualization of thyroid upper pole Limited motion of instrument through bilateral working ports
Sharp dissection	Deeper dissecting plane close to the strap muscle (alternative: subfascial dissection) Compression the bleeder immediately to avoid CO_2_ entering the vessel	Skin burn Anterior jugular vein laceration CO_2_ embolism

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
