# Peer review of "Working Space Creation in Transoral Thyroidectomy: Pearls and Pitfalls"

_cancers, 2022, doi:10.3390/cancers14041031_

Round 1
Reviewer 1 Report
In their study, Tsung-Jung Liang et.al provide a comprehensive insight into the novel approach to the thyroid surgery i.e transoral thyroidectomy
My commentaries:
- What are the inclusion/exclusion criteria for the operation (also, what kind of diagnosis qualifies the patient for the operation)?
- Are there any limitations concerning the tumor size/struma size?
- Can authors make a short comment comparing the risk and difficulty level of this surgical approach in comparison to other approaches?
Author Response
Point 1: What are the inclusion/exclusion criteria for the operation (also, what kind of diagnosis qualifies the patient for the operation)?
Response 1:
Thank you for your comments.
The patient selection for transoral thyroidectomy is similar to that for other remote access thyroidectomy, including the inclusion of those with benign lesions ≤ 5 cm and well-differentiated carcinoma ≤ 2 cm. Larger tumors may be difficult to manipulate in the narrow working space and complete specimen retrieval without capsule disruption may not be feasible through the small oral incision. The integrity of the specimen is very important when operating on patients with suspected malignancy because tissue fragmentation would interfere with the pathological examination. Some surgeons suggest extracting the large specimen with an additional trocar in the axilla.
In the top-down view of transoral thyroidectomy, the lymph nodes at level VI can be clearly identified and central compartment dissection can be performed. However, lateral neck lymph node dissection, especially at level II, is extremely challenging via the transoral approach. Thus, patients with preoperative imaging evidence of lateral neck involvement may be excluded or operated only by experienced surgeons. Other exclusion criteria contain ongoing oral infection and poorly differentiated cancer. Operation in patients with prior neck surgery, previous head and neck irradiation, or extensive thyroiditis are technically demanding and are relatively contraindicated.
Changes in the text:
We have added a paragraph regarding the inclusion and exclusion criteria in the “Basic concepts” section (see page 2, lines 49-66)
Point 2: Are there any limitations concerning the tumor size/struma size?
Response 2:
Thank you for your comments.
As mentioned above, larger tumors are difficult to manipulate, and the specimen cannot be retrieved without fragmentation. The optimal tumor size for transoral thyroidectomy is as follows: benign nodules ≤ 5 cm and malignant nodules ≤ 2 cm
Changes in the text:
We have added a paragraph regarding the tumor size in the “Basic concepts” section (see page 2, lines 49-57)
Point 3: Can authors make a short comment comparing the risk and difficulty level of this surgical approach in comparison to other approaches?
Response 3:
Thank you for your comments.
Previous studies have demonstrated a short learning curve for transoral thyroidectomy, with fewer than 20 cases required for skill acquisition. The learning curve for transoral thyroidectomy is shorter than that for other endoscopic approaches, such as the transaxillary and retroauricular approaches. The symmetric port placement and identical midline approach as open surgery may be beneficial for surgeons to rapidly gain proficiency in transoral thyroidectomy.
Nevertheless, rare but severe complications did occur in transoral thyroidectomy. Surgeons who are interested in using this novel technique should be well-prepared and receive in-depth training before working on their first case.
Changes in the text:
We have added a paragraph for comparing the risk and difficulty level of transoral approach with other approaches in the “Special Consideration” section (see page 11, lines 341-350)
Reviewer 2 Report
In this article, the authors reported the detailed methods of transoral thyroidectomy, and their advantage, and complications. I read this article with interest.
However, I have the worries of categorization of this article. I think this article is expert opinion rather than the review article.
In this article, I cannot find any trial for the review of previous articles systematically. Overall contents are well written expert opinion.
There are no parts for methods section.
In the current status, the conclusion might be deviated without fair selection of previous articles. This should be considered.
Thus, I suggest this article send the expert opinion.
OR the author should fill in the missing parts to be a review article as rigorous systematic review.
Author Response
Reviewer 2
In this article, the authors reported the detailed methods of transoral thyroidectomy, and their advantage, and complications. I read this article with interest.
However, I have the worries of categorization of this article. I think this article is expert opinion rather than the review article.
In this article, I cannot find any trial for the review of previous articles systematically. Overall contents are well written expert opinion.
There are no parts for methods section.
In the current status, the conclusion might be deviated without fair selection of previous articles. This should be considered.
Thus, I suggest this article send the expert opinion.
OR the author should fill in the missing parts to be a review article as rigorous systematic review.
Response:
Thank you for your comments.
The article conforms to the structure of a narrative review, rather than a systemic review. We aimed to provide a step-by-step summary on working space creation in transoral thyroidectomy. Available evidence from anatomical and clinical studies were selected and complied to explain why and how the technique was performed and its evolving development, so that the content may not merely provide expert opinion but also some concrete-evidence backup.
When evaluating common surgical outcomes, such as blood loss or operative time, the definition is clear, and the data are easily available for writing a systemic review. However, data collection on surgical techniques was more difficult because various minor technical modifications were used in different institutes and the details of the procedure were sometimes missing. We think it may be more suitable to present the detailed technique, its variations, and the underlying rationale. This narrative review provides readers with a comprehensive understanding regarding working space creation in transoral thyroidectomy. Some of them may be inspired to further explore the surgical technique and the operative method could be improved for a better performance.
According to the Instructions for Authors, the structure of a review article in Cancers can be flexible, and method section is not mandatory. The following are some examples:
Thyroid Lobectomy for Low to Intermediate Risk Differentiated Thyroid Cancer
https://www.mdpi.com/2072-6694/12/11/3282/htm
Papillary Thyroid Cancer Prognosis: An Evolving Field
https://www.mdpi.com/2072-6694/13/21/5567/htm
Changes in the text:
We have added a statement regarding the limitation and future perspective of this study in the Conclusion section (see page 12, lines 397-399)
Round 2
Reviewer 2 Report
I have no more comments.